# Antioxidant, Antibacterial, Anti-Inflammatory, and Antiproliferative Activity of Sorghum Lignin (*Sorghum bicolor*) Treated with Ultrasonic Pulses

**DOI:** 10.3390/metabo13030394

**Published:** 2023-03-08

**Authors:** Linda Yareth Reyna-Reyna, Beatriz Montaño-Leyva, Dora Valencia, Francisco Javier Cinco-Moroyoqui, Ricardo Iván González-Vega, Ariadna Thalía Bernal-Mercado, Manuel G. Ballesteros-Monrreal, Mayra A. Mendez-Encinas, Carmen Lizette Del-Toro-Sánchez

**Affiliations:** 1Department of Research and Postgraduate in Food, University of Sonora, Blvd Luis Encinas y Rosales S/N, Col. Centro, Hermosillo 83000, Mexico; 2Department of Chemical Biological and Agricultural Sciences, University of Sonora, Av. Universidad e Irigoyen S/N, Caborca 83600, Mexico; 3Department of Medical and Life Sciences, Cienega University Center (CUCIÉNEGA), University of Guadalajara, Av. Universidad 1115, Lindavista, Ocotlán 47820, Mexico

**Keywords:** lignin, sorghum, ultrasonic pulses, organosolv

## Abstract

This investigation aimed to determine the effect of high-power ultrasonic pulses on the antioxidant, antibacterial, and antiproliferative activity of sorghum (*Sorghum bicolor*) lignin. A lignin yield of 7.35% was obtained using the organosolv method. Additionally, the best conditions of the ultrasonic pulses were optimized to obtain a more significant increase in antioxidant capacity, resulting in 10 min for all treatments, with amplitudes of 20% for DPPH and FRAP, 18% for ABTS, and 14% for total phenols. The effect of ultrasonic pulses was mainly observed with FRAP (1694.88 µmol TE/g), indicating that the main antioxidant mechanism of lignin is through electron transport. Sorghum lignin with and without ultrasonic pulses showed high percentages of hemolysis inhibition (>80%) at concentrations of 0.003 to 0.33 mg/mL. The AB blood group and, in general, all Rh- groups are the most susceptible to hemolysis. Lignin showed high anti-inflammatory potential due to heat and hypotonicity (>82%). A higher antimicrobial activity of lignin on *Escherichia coli* bacteria was observed. The lignins evaluated without sonication and sonication presented higher activity in the cell line PC-3. No effect was observed on the lignin structure with the FT-IR technique between sonication and non-sonication; however, the organosolv method helped extract pure lignin according to HPLC.

## 1. Introduction

Sorghum (*Sorghum bicolor*) is one of the cereals that significantly benefits humans and animals due to its agronomic and nutritional characteristics as food. Consequently, the demand for and production of cereal crops have increased with difficulty and, therefore, the generation of agricultural residues (plant remains once the grains are harvested). In Mexico, sorghum has a planting area of 895,026 ha with a production of 4,012,508 t in 2019, whereas Sonora contributes 312,546 t with a planting area of 18,552 ha [1]. According to the Secretariat of Energy of the Federal Government of Mexico, approximately 10 t of waste originates for each hectare produced, generating about 8.95 million tons per year at the national level, and in Sonora, only 185,520 t originate. This residue mainly comprises cellulose, hemicellulose, and lignin (15–40%) [2]. Lignin is the second most abundant polymer on Earth, after cellulose, and it is an affordable renewable resource with potential industrial use. However, the structural definition of lignin has never been as clear as that of other natural polymers, such as cellulose and proteins, due to the complexity that affects its isolation, compositional analysis, and structural characterization. The problem of a precise definition for lignin is associated with the nature of its multiple structural units, which do not usually repeat themselves regularly since the composition and structure of lignin vary depending on its origin, and the method of extraction or isolation used [3].

Notably, the only non-polysaccharide fibers known chemically are lignins, which are phenolic polymers [4]. In general, lignins are copolymers derived primarily from three basic phenylpropane-monomeric units (monolignols): p-coumaryl alcohol, coniferyl alcohol, and sinapyl alcohol. These monolignols react in the cell wall through oxidation reactions catalyzed by peroxidases (radical intermediates) to form lignin polymers. The distribution percentages of the main monolignols depend on the type of plant. Hydrolysis and oxidation of lignin at high pressures and temperatures produce low molecular weight compounds. These compounds represent a variety of high-added-value chemicals, the most important being a group of phenolic compounds that include vanillin, ferulic acid, syringic acid, eugenol, cresols, catechols, and guaiacol [3]. Phenolics are compounds considered antioxidants (they delay or prevent oxidation) that neutralize free radicals that cause cell damage and the development of chronic-degenerative diseases [5]. Evidence has shown that lignin extracted from sorghum residues has a high antioxidant capacity (75% inhibition of the DPPH radical) [6]. The antioxidant capacity, on many occasions, has also been related to other biological activities, such as antimicrobial and anticancer properties [7]. However, these activities have not been determined in sorghum residues. 

However, the organosolv method (extraction with organic solvents) is a method in which purer lignins are obtained with low polydispersity, specific molecular weights, and high phenolic composition [6]. However, it has been shown that applying high-power pulses increases the permeability of biological membranes, contributing to better extraction of different classes of therapeutically active compounds and affecting the bioactivity of the compounds [8] due to the process called cavitation. Cavitation consists of mechanical waves that require an elastic medium to spread, creating a longitudinal displacement of the particles, resulting in a succession of compression and rarefaction phases. Cavitation bubbles form voids in the middle whenever the rarefaction cycle is strong enough. When these bubbles collapse on the surface of a solid material, the high pressure and temperature released generate microjets that are directed toward the solid surface, destroying the cell walls, allowing the release of the compounds of interest and thus influencing the increase of its biological activity [9,10]. The objective of this research was to evaluate the effect of high-power pulses on antioxidant, antibacterial, and anticancer activity and the structure of sorghum lignin.

## 2. Materials and Methods

### 2.1. Reagents

Reagents, such as ABTS [2,2′-azino-bis-(3-ethylbenzothiazoline-6-sulfonic acid], AAPH [2,2′-Azobis(2-methylpropionamidine) dihydrochloride], DPPH [2,2-diphenyl-1-picrylhydrazyl], Trolox [6-hydroxy -2,5,7,8-tetramethylchroman-2-carboxylic acid], Folin–Ciocalteu, TPTZ [2,4,6-Tripridil-s-triazine], HPLC-solvents, bovine serum albumin (BSA), and Tris-HCl biological buffer, were purchased from Sigma-Aldrich Co. (San Luis, MO, USA). All other chemicals and solvents were of the highest commercial grade.

### 2.2. Extraction of Lignin by the Organosolv Method

The method described by Pan et al. [11] was followed. Agricultural residues of the ‘Argos’ sorghum variety, harvested during the 2021 agricultural cycle in the Yaqui Valley (geographic coordinates between 27°00′ and 28°00′ N and 109°30′ and 110°37′ W in the state of Sonora, Mexico), were used. An amount of 10 g of sample was added to a solution consisting of formic acid, acetic acid, and water (30:60:10, *v*/*v*/*v*) to prepare a 1:20 (*w*/*v*) solid-to-solution ratio mixture. The mixture was heated in a water bath for 4 h at 80 °C with constant agitation. Two phases were obtained: a solid one consisting of cellulose and a liquid one that contained traces of solvents, hemicellulose, and lignin. This liquid phase was concentrated in a rotary evaporator, adding three volumes of 95% ethanol at the end, and allowed to rest for 12 h at room temperature to precipitate the carbohydrates. Afterward, the mixture was filtered through Whatman 47 mm paper filters to remove the precipitated carbohydrates. The procedure was repeated three times. The dark brown liquid obtained after filtering was concentrated, which contained the Organosolv lignin. The concentrate was washed by taking 1 ml of the Organosolv lignin and combining it with 47 mL of distilled water. The mixture was centrifuged at 9200× *g* for 10 min. The supernatant was decanted, and one more milliliter of the sample was added, repeating the washing twice. Finally, the Organosolv lignin was recovered and allowed to dry at room temperature to determine its yield.

### 2.3. Determination of Antioxidant Activity

The antioxidant activity of lignin was determined by the methods of ABTS (2,2′-azinobis(3-ethylbenzothiazoline)-6-sulfonic), DPPH (Radical 2,2-diphenyl-1-picrilhydrazil), and FRAP (Ferric Reducing Antioxidant Power).

#### 2.3.1. ABTS

The antioxidant activity determined by the ABTS method was carried out following the procedure of Re et al. [12]. A solution of the radical ABTS·+ (3.86 mg/mL) was combined with 88 µL of 2.45 mM potassium persulfate. The mixture was allowed to rest for 16 h in the dark. Subsequently, 1 mL of this solution was collected and combined with ethanol until an absorbance of 0.7 ± 0.02 at 734 nm was reached. Once set up, 20 μL of the sample was mixed with 270 μL of the radical solution and allowed to stand for 30 min at room temperature in the dark. After this time, the absorbance was measured in a 96-well microplate reader (Thermo Fisher Scientific Inc., Multiskan GO, New York, NY, USA). The measurements were performed in triplicate. The results were expressed as micromoles of Trolox equivalents per gram of sample (µmol TE/g).

#### 2.3.2. DPPH

The antioxidant activity was determined through a colorimetric method based on the decrease in absorbance of DPPH (free radical) in the presence of the lignin extract following the procedure of Loarca-Piña et al. [13]. The DPPH stock solution was diluted with methanol to give an absorbance of 0.7 ± 0.02. To 20 µL of the lignin extract, 200 µL of a freshly prepared DPPH solution was added and incubated for 30 min in the dark. The absorbance was measured at 515 nm in a microplate spectrophotometer (Thermo Scientific Multiskan Go, New York, NY, USA). The measurements were performed in triplicate. The results were expressed as micromoles of Trolox equivalents per gram of sample (µmol TE/g).

#### 2.3.3. FRAP

FRAP was measured by the reducing power of the ferric to ferrous ion according to the methodology of Benzie and Strain [14]. Initially, individual sodium acetate buffer solutions (300 mmol/L, pH 3.6) were prepared: iron-TPTZ complex with FeCl_3_ 6H_2_O (20 mmol/L) and a solution of TPTZ in HCl (40 mmol/L). Subsequently, the FRAP solution was prepared in a ratio of 10:1:1 from the previously made solutions (Buffer: FeCl_3_·6H_2_O: TPTZ·HCl). An aliquot of 280 µL of the FRAP solution was mixed with 20 µL of the sample, and the absorbance was measured at 638 nm every 10 min for 60 min. The measurements were performed in triplicate. The results were expressed as µmol TE/g sample from a Trolox curve (4000 at 15.6 µmol).

### 2.4. Total Phenolic Content

Total phenols were determined using the method of Ainsworth and Gillespie [15] with some modifications. An aliquot of 10 µL (1 mg/mL of DMSO) of the extract was combined with 25 µL of 1 N Folin–Ciocalteu reagent, 25 µL of sodium carbonate (20% Na_2_CO_3_), and 140 µL of milli-Q water. The mixture was placed in the wells of a 96-well microplate and incubated and protected from light at room temperature for 30 min. The absorbance was measured at 760 nm in a microplate reader (Thermo Scientific Multiskan Go, New York, NY, USA). The measurements were performed in triplicate. The results were reported as mg gallic acid equivalents/g dry extract (mg GAE/g) based on a gallic acid standard curve.

### 2.5. Application of Ultrasonic Pulses to Sorghum Lignin

Tubes with 10 mL of the sample at a concentration of 1 mg/mL were prepared for the application of ultrasonic pulses (Generator ultrasonic pulses Branson Digital Sonifier QSonica, LLC., Brookfield, CT, Middleboro, Massachusetts, USA) according to the combinations provided by the Composite Central Design (Table 1) to optimize the best conditions for obtaining higher antioxidant capacity and total phenols in sorghum lignin. The statistical package JMP version 15.0 (SAS Institute Inc., Cary, North Carolina, USA) was used. During the application of the pulses, 1 min of rest was left after each minute of the pulse applied until the desired time was completed. Subsequently, measurements of antioxidant capacity and quantification of total phenols were performed.

### 2.6. Determination of Antihemolytic Activity

The assay was performed for each sample separately, following the technique described by González-Vega et al. [16] using AAPH. Blood samples were obtained from healthy volunteers older than 18 with prior informed consent. Approximately 5 mL of blood samples (different blood types A+, B+, O+, AB+, A−, O−) were taken in heparin tubes. A 2% solution of erythrocytes was prepared, in which 1 mL of blood was resuspended in 3 mL of physiological solution, homogenized by inversion, and centrifuged at 2000× *g* for 10 min. This step was performed in triplicate to wash the sample. Subsequently, the supernatant was removed. One hundred microliters of washed erythrocytes were collected and suspended in 5 mL of physiological solution. In five Eppendorf tubes, the erythrocyte suspension was mixed with each sample (sorghum lignin without sonication and with sonication). Then, AAPH (0.1085 g/mL, pH 7.4) was added as follows: Samples (100 μL of erythrocyte suspension + 100 μL of AAPH + 100 μL of lignin sample); positive control (100 μL of erythrocyte suspension + 100 μL of saline solution + 100 μL of AAPH); and negative control (100 μL of erythrocyte suspension + 200 μL of physiological solution).

The tubes were placed at 37 °C in a water bath for 3 h with constant shaking. After that time, 1 mL of physiological solution was added to each tube and centrifuged at 2000× *g* for 10 min. Three hundred microliters of supernatant was collected and placed in the 96-well microplate wells. Absorbance was measured at 540 nm in a microplate reader (Thermo Scientific Multiskan Go, New York, NY, USA). The percentage of hemolysis inhibition (%HI) was calculated from the following Equation (1):(1)$$\%HI=\left(\frac{\mathrm{AAPH}1-\mathrm{HS}}{\mathrm{AAPH}1}\right)\;\times \;100$$
where AAPH1 = optical density of hemolysis caused per AAPH. HS = optical density of the hemolysis inhibition by each treatment.

### 2.7. Determination of Anti-Inflammatory Activity

The hypotonicity-induced hemolysis and heat-induced hemolysis of human red blood cells methods were used to assess membrane stabilization using the technique described by Shinde et al. [17]. The AB blood type could not be used for this technique due to its scarcity.

#### 2.7.1. Heat-Induced Hemolysis

A solution of 150 μL of erythrocyte suspension and 100 μL of each lignin sample in different solutions from 0.01 to 10 mg/mL was prepared and incubated at 55 °C for 30 min. Then, 1 mL of physiological solution was added and centrifuged at 2000× *g* for 10 min. Next, 300 µL of the supernatant was taken from each tube and placed into a 96-well plate. The absorbance was read at 560 nm in a microplate reader (Thermo Scientific Multiskan Go, New York, NY, USA).

#### 2.7.2. Hemolysis Induced by Hypotonicity

An aliquot of 50 μL of erythrocyte suspension was mixed with 100 μL of each lignin sample in different solutions from 0.01 to 10 mg/mL, combined with 100 μL of saline solution plus 200 μL of the hypotonic solution, and incubated at 37 °C for 30 min. Subsequently, 850 μL of the physiological solution was added and centrifuged at 2000× *g* for 10 min. After that, 300 µL of the supernatant was taken from each tube and added to a 96-well plate. It was read at an absorbance of 560 nm in a microplate reader (Thermo Scientific Multiskan Go, New York, NY, USA).

### 2.8. Antibacterial Activity

For this assay, strains ATCC 25922 (*Escherichia coli*) and ATCC 25923 (*Staphylococcus aureus*) were obtained from the microbiology laboratory of the Caborca campus of the University of Sonora, Mexico [18]. Fresh 14–16-hour-old colonies were prepared on Müeller Hinton agar from each strain. Subsequently, the colonies were taken and mixed in sterile saline solution (0.85%) using the microdilution method until they reached 0.95 ± 0.05 of optical density at 630 nm according to the McFarland scale. A 40 mg/mL stock solution of lignin was made in sterile DMSO. Subsequently, 1:2 solutions were made in the 800 to 100 µg/mL concentration range. A 200 µL aliquot of each sample was taken in triplicate wells of 96-well microplates. Wells containing medium + bacteria, medium without bacteria, medium + antibiotic gentamicin (12 µg/mL), and medium + solvent were used as controls. A volume of 15 µL of the preparation with previously standardized bacteria was added to all the wells, incubated at 36 °C for 30 min, and the optical density was read at 630 nm. Bacterial development curves were made from these measurements by plotting time against optical density.

### 2.9. Evaluation of Antiproliferative Activity in Cancer Cell Lines

To evaluate the antiproliferative effect of lignin on cancer cell lines, L-929 (mouse fibroblast cell line), HeLa (cervical adenocarcinoma), and PC3 (human prostate cancer) cell lines were provided by the University of Sonora, Campus Caborca. The cell lines were grown in Dulbecco’s Modified Eagle’s medium (DMEM) with 5% fetal bovine serum at 37 °C, with an atmosphere of 5% CO_2_ and 80–90% RH in a Thermo Scientific incubator (HERACELL VIOS 160i).

The antiproliferative activity was determined using the tetrazolium salt reduction method (MTT, 3-(4,5-dimethyl-2-thiazoyl)-2,5-diphenyltetrazolium bromide, Sigma-Aldrich Co., St. Louis, MO, USA) according to Mosmann [19] with slight modifications made by Hernandez et al. [20]. Briefly, cells were seeded (10,000 cells per well, 50 µL) in 96-well plates and incubated for 24 h at standard culture conditions (37 °C and 5% CO_2_). After incubation, aliquots of DMEM (50 µL) containing different concentrations of the samples (200 to 0 µg/mL) were added into each well and incubated for 48 h. DMSO was used as a dissolvent control. In the last 4 h of incubation, 10 µL of MTT (5 mg/mL) was added to each well. Cell viability was assessed by the ability of metabolically active cells to reduce tetrazolium salt to colored formazan compounds. The resulting formazan crystals were dissolved with acidic (0.3%) isopropyl alcohol. The absorbance was measured in an ELISA plate reader (Multiskan EX, ThermoLabSystem, Waltham, MA, USA) using a test wavelength of 570 nm and a reference wavelength of 650 nm. The antiproliferative activity was reported as IC_50_ values obtained by linear regression analysis.

### 2.10. Lignin Analysis by Fourier Transform Infrared Spectroscopy (FT-IR)

Lignin samples were analyzed using Thermo Scientific Nicolet iS-50 equipment (Madison, WI, USA). All runs were carried out between 4000 and 500 cm^−1^ using a total accessory reflection (ATR) attenuated with an ATR diamond crystal using 64 scans with 4 cm^−1^ resolution.

### 2.11. Molecular Weight Determination by Size Exclusion Chromatography (HPLC-SEC)

The molecular mass of lignins was determined using size exclusion high-pressure liquid chromatography (SE-HPLC) equipment (Agilent 1260 Infinity Quaternary LC System, mod. 240; Palo Alto, CA, USA) equipped with an Agilent 1260 Infinity refractive index detector. Separation by size was performed on a TSK-GEL Alpha-300 column (30 cm × 7.8 mm, 7 μm particle size; Tosoh Bioscience, Stuttgart, Germany) calibrated with polystyrene standards (Sigma-Aldrich, St. Louis, MO, USA) ranging from 266 to 70,000 Da in molecular weight. Lignin samples (25 mg) were placed in microtubes and combined with 1 mL of 0.1 M lithium chloride (LiCl) in dimethylformamide (DMF). The samples were vortexed for 30 min, left under constant agitation for 24 h, filtered through 0.45 µm acrodiscs, and placed in vials for analysis. The injection volume was 20 μL, and the column temperature was maintained at 40 °C. The solvent used for elution was 0.1 M LiCl in DMF at a flow rate of 1 mL/min. 

### 2.12. Statistical Analysis

Data were subjected to linear and non-linear regression and one-way analysis of variance (ANOVA) by comparing means with the Tukey test (*p* < 0.05). All data were analyzed using the statistical program JMP software v15 and R-Studio software v1.1.463 for Mac. The study was conducted under controlled conditions, with a minimum of three repetitions for each analysis.

## 3. Results and Discussion

### 3.1. Yield of Lignin Obtained from Sorghum Residues

The results of the lignin yield from sorghum residues through the Organosolv method are shown in Table 2. The lignin extraction yield was 7.35%. There are few studies related to the extraction of lignin from sorghum; there is only one study by Gaspar García [6] where a yield of approximately 8% was obtained using the same methodology. Organosolv processes involve the removal of lignin from lignocellulosic biomass in plant matter and have been developed in support of the pulping industry. Organosolv is practically carried out by the solvents used to separate hemicellulose, cellulose, and lignin. Therefore, the advantage of extracting lignin with the Organosolv method is that purer lignins can be obtained, making it ideal for direct use and conversion into high-value chemicals.

On the other hand, the Organosolv process is valuable for producing sulfur-free lignin from biomass. To date, there is no commercial Organosolv lignin on the market. Most efforts are still at the laboratory scale [21]. However, this methodology could help obtain lignin from sorghum residues, providing another alternative for using and applying this agroindustrial waste. However, the recovery and recycling of the solvents used must be economically competitive.

### 3.2. Optimization of Sorghum Lignin Subjected to High-Power Ultrasonic Pulses

The results obtained from the antioxidant capacity of ABTS, DPPH, FRAP, and total phenols according to the Central Composite Design matrix are shown in Table 1. Figure 1 shows the response surface graphs of the optimizations for DPPH (Figure 1A), ABTS (Figure 1B), FRAP (Figure 1C), and total phenols (Figure 1D). In this figure, an increase in the antioxidant activity of the evaluated methods can be observed, mainly in the FRAP method, since results above 1600 µmol TE/g sample were observed, indicating that the mechanism of action of lignin may be by electron transfer, since the FRAP study measures the reducing power. In a study by Cheng et al. [22], lignin samples showed FRAP activity in the range of 2.18 mmol/g to 3.41 mmol/g, and they determined that the observed FRAP value could be due to the phenolic hydroxyl group.

The DPPH radical method has been used constantly to determine the antioxidant capacity of lignin obtained from different sources. According to the references, the antioxidant activity of lignin depends on the method of obtaining it and the purification process, as well as its phenolic hydroxyl groups, molecular weight, and polydispersity [11]. However, the values analyzed by the ABTS assay were higher than those calculated by the DPPH assay. This could be explained by the difference in the affinity of the compounds due to their polarity. For instance, ABTS has an affinity for hydrophilic and hydrophobic compounds [23,24]. In contrast, DPPH has less affinity for compounds containing aromatic rings with hydroxyl groups only and more affinity for lipophilic compounds [25].

Table 3 and Table 4 show a summary of the optimized conditions, where all analyses were at 10 min with amplitudes of 20% for DPPH and FRAP, 18% for ABTS, and 14% for total phenols. Other studies have been conducted to evaluate the biological activities of different materials to which ultrasonic pulses were applied. For instance, the use of specific conditions of ultrasonic pulses (200 W, 20 kHz; 55 °C; 20 min) on marine algae increased the antibacterial (agar diffusion method) and anticancer activities (MTS assay) [26].

### 3.3. Antihemolytic Activity

#### 3.3.1. Effect of Blood Groups on the Inhibition of Free Radicals Induced by AAPH Adding Sorghum Lignin with and without Pulses

The results obtained from the antihemolytic activity are shown in Figure 2. There was no significant difference between groups A+, B+, and O+ at concentrations from 0.003 to 0.03 mg/mL in lignins without ultrasonic pulses and with pulses of 10 min at 14 and 18 MHz. In the sample of lignin extracted using 20 MHz, only the concentration of 0.003 mg/mL showed no differences between the same groups. The AB+ group generally presented the lowest percentage of hemolysis, and it was observed that increasing the lignin concentration increased this percentage slightly. At the highest lignin concentration in all cases, a decrease in inhibition percentage was observed, especially in groups A, B, and O+. The O+ blood group with lignin sonicated at an 18 MHz amplitude (concentration of 0.33 mg/mL) and that at 20 MHz (concentration of 0.003 to that of 0.33 mg/mL) was the best group with less oxidative damage, which resulted in high percentage values of hemolysis inhibition.

The study carried out by Shabbir et al. [27] studied fractions of *M. royleanus*. It showed that the lysis of the erythrocytes increased concomitantly with the concentration of the extract, which is comparable with sorghum lignin since, at a higher concentration, it presents more significant lysis of the erythrocytes. At a lower concentration, it protects the membrane from this. Therefore, two possible mechanisms are proposed by which lignin can act by inhibiting erythrocyte lysis; one is that lignin protects the erythrocyte from free radicals in the cell membrane. Another possible mechanism is that it stabilizes the free radicals generated by AAPH before they reach the erythrocyte.

#### 3.3.2. Effect of Rh on the Inhibition of Free Radicals Induced by AAPH with Lignin with and without Pulses

Figure 3 shows the effect of Rh on the inhibition of free radicals. The results indicated that the degree of inhibition depends on the lignin concentration, both with ultrasonic pulses and without pulses. In general, Rh-negative groups were more susceptible to hemolysis, mainly in group O. However, it can also be observed that, depending on the lignin concentration, in some cases, the Rh- groups presented higher hemolysis percentage values than the Rh-positive ones. A possible mechanism for that would be that lignin shows some affinity for the protein found in erythrocytes interacting with it; therefore, in that way, hemolysis might be prevented more in the Rh+ groups than in the Rh- ones.

### 3.4. Anti-Inflammatory Activity

#### 3.4.1. Effect of Blood Groups on Anti-Inflammatory Activity by Heat in Lignin with Pulsed and Non-Ultrasonic Pulses

The results of the anti-inflammatory activity by heat are shown in Figure 4. High percentage inhibition values of heat-induced hemolysis (>80%) were determined in all treatments, except lignin extracted with ultrasonic pulses at 10 min and 20 MHz amplitude, where a decrease was observed mainly in concentrations from 0.003 to 0.33 mg/mL. A reduction in blood group A+ was also observed at a concentration of 3.33 mg/mL in lignin without sonication. From what is known, membrane stabilization has been proposed as a critical mechanism in the anti-inflammatory activity of various medicinal plants. In this case, lignin has anti-inflammatory potential in the different types of blood used with sonicated and non-sonicated lignin; a possible explanation could be the interaction with membrane proteins by sorghum lignin.

#### 3.4.2. Effect of Blood Groups on Anti-Inflammatory Activity by Lignin Hypotonicity with Ultrasonic Pulses and without Ultrasonic Pulses

Figure 5 shows the results obtained for the anti-inflammatory activity. The results indicated that most of the lignins in different concentrations and with different blood groups have a high percentage of potential for hemolysis induced by hypotonicity and do not have significant differences between the blood groups of each lignin evaluated. It is considered that the erythrocyte membrane is analogous to the lysosomal membrane, and its stabilization by plant extracts is regarded as a good indicator of its anti-inflammatory activity. A possible explanation for the mechanism by which the lignin extract induces resistance to erythrocyte osmotic lysis could be the increased retention of intracellular solutes or the interaction with membrane proteins.

### 3.5. Antibacterial Activity

The use of plant extracts and phytochemicals is of great importance in therapeutic treatments. Several scientific studies have been conducted in different countries to demonstrate such efficiency in recent years. Many plants have been used because of their antimicrobial characteristics attributed to the compounds synthesized by their secondary metabolism. For the bactericidal activity of these extracts, *E. coli* ATCC 25922 and *S. aureus* ATCC 25923 strains were implemented as Gram-negative and Gram-positive bacterial models, respectively. Both species are of clinical importance, as they are associated with several infectious processes in humans, in addition to the more frequent reports of clinical isolates of multidrug-resistant *S. aureus* and *E. coli*. The evaluation of lignin organosolv to inhibit the growth of *E. coli* and *S. aureus* is shown in Figure 6. The concentrations used for the antibacterial evaluation of the extract were 800, 400, 200, and 100 µg/mL. All lignins evaluated at a concentration of 800 ug/mL decreased the percentage of viability of both bacteria; however, this effect was more remarkable in the case of Gram-negative bacteria. This selective action may be due to differences in the chemical composition of the cell walls.

### 3.6. Antiproliferative Activity of Lignin on Cell Lines

To evaluate the antiproliferative activity of lignin organosolv, two widely studied cancerous cell lines (HeLa and PC-3) and one non-cancerous (L-929) were selected as models for the study. The effect of lignin on cell proliferation is shown in Figure 7. In interpreting the results obtained, the activity of the extracts was considered high if the IC_50_ was ≤30 µg/mL, average between 30–100 µg/mL, and low >100 µg/mL [28,29]. The concentrations evaluated were 100, 200, 400, and 800 µg/mL. Table 5 shows the IC_50_ values for the samples evaluated in such cell lines. The order of antiproliferative activity of the extracts was as follows: in the cell line for LCS, PC3 > HeLa, with IC_50_ values of 126.52 ± 4.83 and 252.07 ± 28.56 µg/mL respectively, and for L1 PC3 > HeLa, with IC_50_ values of 171.85 ± 18.91 and 284.90 ± 23.84 µg/mL respectively. Regarding L2 and L3, in the same way, the PC3 cell line presented better antiproliferative activity; however, based on the parameters established for interpreting the values, the lignins evaluated showed low antiproliferative activity in the cell lines.

### 3.7. Analysis by Infrared Spectroscopy with Fourier Transform (FT-IR)

Figure 8 shows the infrared spectrum of sorghum lignin with and without treatment by high-power ultrasonic pulses. Additionally, Table 6 shows the absorption bands or peaks corresponding to the vibrations of lignin’s most important chemical species. It can be observed that the spectra of the lignin samples treated with ultrasonic pulses L1 (10 min/20 MHz), L2 (10 min/18 MHz), and L3 (10 min/14 MHz) did not present differences concerning the untreated sample LCS (complete sorghum lignin). A band observed between 3276 cm^−1^ is typical of hydroxyl groups (O–H stretching) in phenolic and aliphatic structures. In contrast, the bands found at 2918 and 2850 cm^−1^ were caused by sp3 C-H stretching on methyl groups (–CH_3_), methylene groups (=CH_2_/–CH_2_–), and methoxy groups (–OCH_3_). The same peaks can also be assigned to C-H stretching in aldehydes [30].

It is important to highlight the presence of bands at 1605 and 1516 cm^−1^ related to C=C stretching of the aromatic ring and at 1440 cm^−1^ related to C-H strains in organosolv lignin. The appearance of these bands suggests that the aromatic core of the sorghum lignins was maintained even using acid solutions and high temperatures. The C-H stretch of the guaiacil (G) units is shown at 1295 cm^−1^. The band at 1167 cm^−1^ is characteristic of the presence of lignin units H, G, and S [31]. This indicates that the sorghum lignin extracted by the organosolv method presents an HGS-type structure in this study. These results agree with those found by de Sousa Nascimento et al. [32], who extracted lignin from banana peel by the organosolv method and observed a band at 1169 cm^−1^ characteristic of HGS-type lignins. In the same study, a subtle peak was observed at 794 cm^−1^ related to out-of-plane C-H at positions 2, 5, and 6 of the guayacil units. Similarly, this band appeared in the spectra of lignin organosolv from sorghum. The sorghum lignins extracted by the organosolv method showed the presence of guayacil and syringyl groups, demonstrating that lignins can be extracted with this process while preserving most of their original structure.

Carbonyl groups are generated during acid hydrolysis signaling at 1727 cm^−1^. In the same way, during treatment with organic acids, a certain degree of oxidation is generated in the lignin structure. The band observed at 1370 cm^−1^ shows that this oxidation occurs mainly in the gamma position in the side chain of the lignin structure [11]. The infrared spectra of sorghum lignins extracted by the organosolv method were similar, with no change in peak characteristics or relative intensities. This suggests that, under the conditions in this work, high-frequency ultrasonic pulses do not promote changes in the chemical structure of lignin.

### 3.8. Analysis by High-Performance Chromatography (HPLC)

High-performance liquid chromatography (HPLC) was used to analyze the sorghum lignin with and without ultrasonic pulses. According to the chromatograms obtained (Figure 9), it was possible to verify the presence of lignin since symmetrical peaks were observed with well-defined crests. In addition, other signals with less height indicate that other compounds may have been generated during the oxidation of the lignin.

In the LCS sample, a peak is observed between 8.5–14 min, a region that corresponds to the smallest molecular weights. A shoulder was also noted in this peak at a retention time of 6.5–8.3 min. It is considered that this sample presents polydisperse behavior. However, if the majority band is considered, it can be said that the approximate molecular weight is 2400 Da. These results agree with Zhang et al. [33], who compared different molecular weights of lignins treated with other methods.

The samples to which the ultrasonic pulses were applied showed a slight behavior change, and the shoulder was less noticeable. However, considering the maximum peak, there were no changes in the molecular weight of the samples. This is consistent with the FTIR results, which show that the pulses caused no structural changes.

Table 7 shows the lignins evaluated with retention times and molecular weights obtained, where no differences were shown. Thus, we can conclude that high-power ultrasonic pulses did not affect the lignin structure. Organosolv-type lignins have a generally low molecular mass and polydispersity compared to other lignin types [34], which coincides with our results since we obtained lignins with these characteristics.

## 4. Conclusions

It was possible to determine the optimized conditions of the ultrasonic pulses with the greatest increase in antioxidant capacity. All analyses were conducted at 10 min, with amplitudes of 20% for DPPH and FRAP, 18% for ABTS, and 14% for total phenols. High-power ultrasonic pulses increased the antioxidant capacity of sorghum lignin, mainly in the FRAP method, indicating that the possible mechanism of the antioxidant action of lignin is through electron transfer. Sorghum lignin with and without ultrasonic pulses showed high percentages of hemolysis inhibition (>80%), with group AB being the most susceptible to hemolysis. In the effect of Rh in inhibiting free radicals induced by AAPH with lignin at different times and amplitudes, an increase in activity was seen in Rh+. Therefore, Rh– was more susceptible to hemolysis. High anti-inflammatory potential due to heat and hypotonicity (>82% both) of sorghum lignin was shown in the different blood groups. Concerning the antibacterial activity of the evaluated lignins, better activity was observed against *E. coli* bacteria (Gram-negative). The lignins evaluated without sonication and with sonication presented greater activity in the cell line PC-3. No differences were observed between the FT-IR of the analyzed lignins and their molecular weight; therefore, more studies are required to analyze the effect of high-power ultrasonic pulses on the lignin structure.

## Figures and Tables

**Figure 1 metabolites-13-00394-f001:**
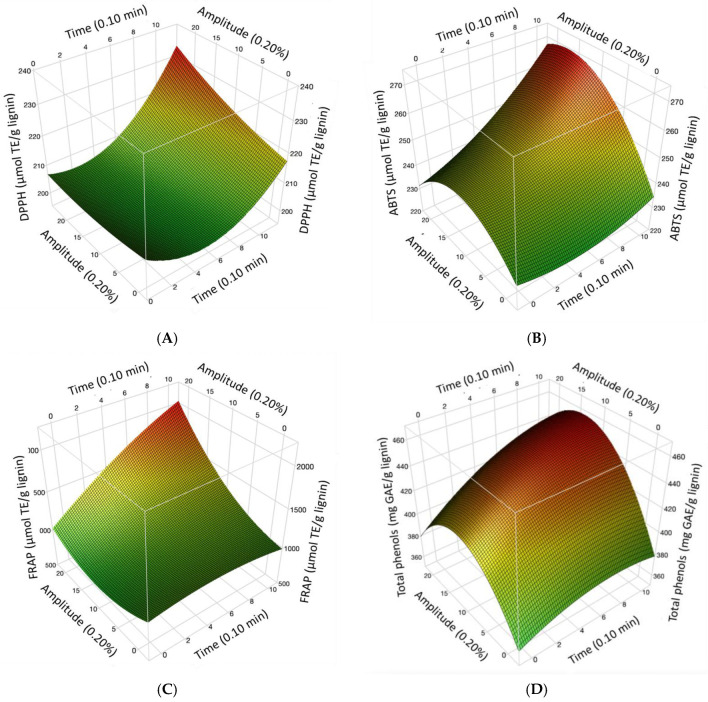
Response surface: Effect of ultrasonic pulses on the antioxidant capacity by (**A**) DPPH, (**B**) ABTS, (**C**) FRAP, and (**D**) Total phenols in sorghum lignin.

**Figure 2 metabolites-13-00394-f002:**
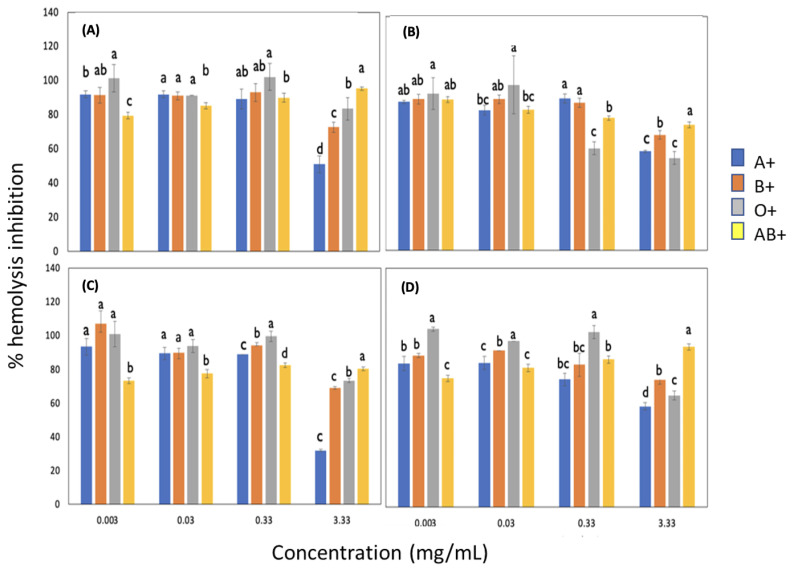
Effect of blood groups with different Rh in inhibiting free radicals induced by AAPH in lignin with and without pulses. (**A**) Lignin with 0/0 time (min)/amplitude (MHz), (**B**) Lignin with 10/14 time (min)/amplitude (MHz), (**C**) Lignin with 10/18 time (min)/amplitude (MHz), (**D**) Lignin with 10/20 time (min)/amplitude (MHz). Different letters indicate significant difference (*p* < 0.05).

**Figure 3 metabolites-13-00394-f003:**
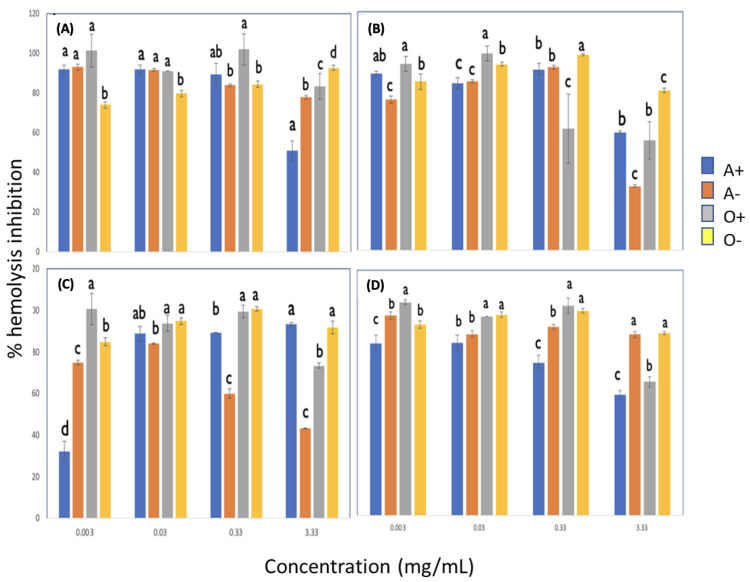
Effect of Rh on inhibiting free radicals induced by AAPH with lignin and without pulses. (**A**) Lignin with 0/0 time (min)/amplitude (MHz), (**B**) Lignin with 10/14 time (min)/amplitude (MHz), (**C**) Lignin with 10/18 time (min)/amplitude (MHz), (**D**) Lignin with 10/20 time (min)/amplitude (MHz). Different letters indicate significant difference (*p* < 0.05).

**Figure 4 metabolites-13-00394-f004:**
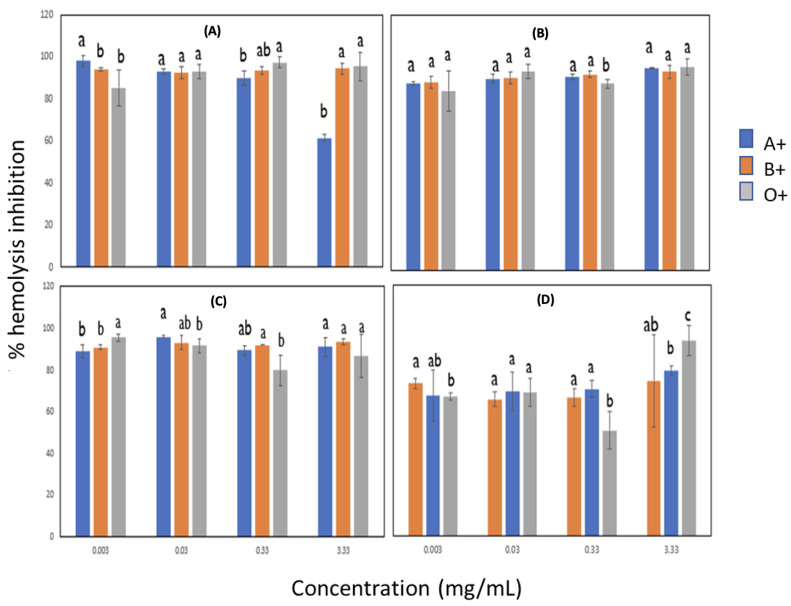
Effect of blood groups on anti-inflammatory activity by heat in lignin with pulsed and non-ultrasonic pulses. (**A**) Lignin with 0/0 time (min)/amplitude (MHz), (**B**) Lignin with 10/14 time (min)/amplitude (MHz), (**C**) Lignin with 10/18 time (min)/amplitude (MHz), (**D**) Lignin with 10/20 time (min)/amplitude (MHz). Different letters indicate significant difference (*p* < 0.05).

**Figure 5 metabolites-13-00394-f005:**
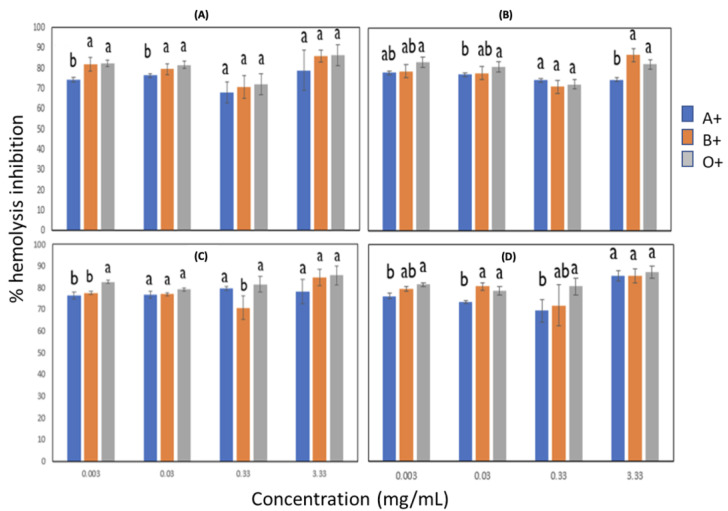
Effect of blood groups on anti-inflammatory activity due to hypotonicity in lignin with and without ultrasonic pulses. (**A**) Lignin with 0/0 time (min)/amplitude (MHz), (**B**) Lignin with 10/14 time (min)/amplitude (MHz), (**C**) Lignin with 10/18 time (min)/amplitude (MHz), (**D**) Lignin with 10/20 time (min)/amplitude (MHz). Different letters indicate significant difference (*p* < 0.05).

**Figure 6 metabolites-13-00394-f006:**
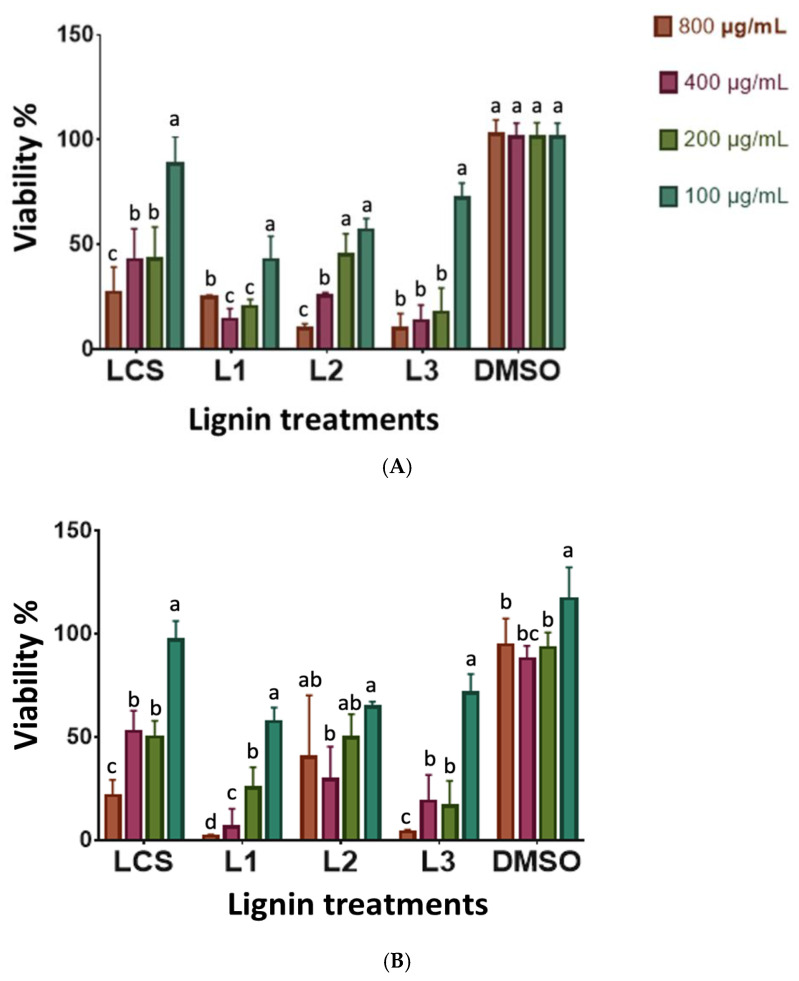
Antimicrobial activity of sorghum lignin extracts without treatment (LCS) and with different treatments (L1) Lignin treatment time/amplitude-10/20. (L2) Lignin treatment time/amplitude-10/18. (L3) Lignin treatment time/amplitude-10/14 vs. growth curve of (**A**) *S. aureus* ATCC 25923 and (**B**) *E. coli* ATCC 25922. DMSO was used as the solvent control. The results are representative of at least three independent experiments. Different letters indicate significant difference (*p* < 0.05).

**Figure 7 metabolites-13-00394-f007:**
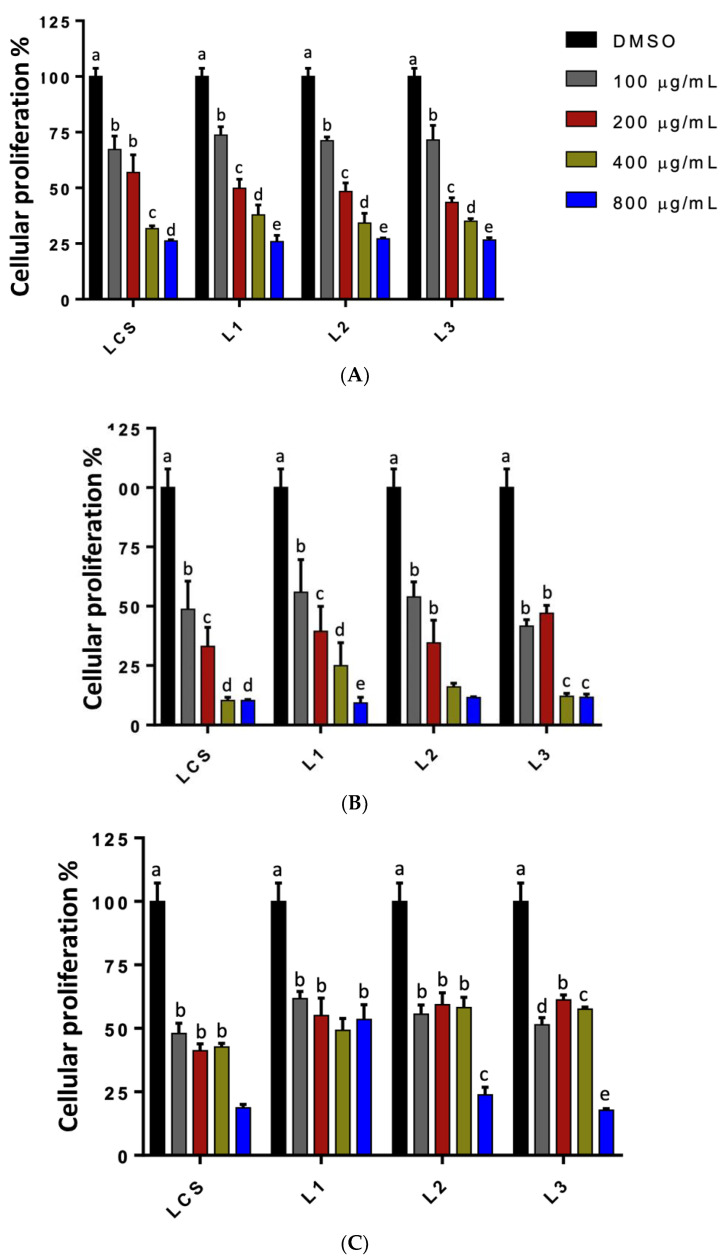
Antiproliferative activity on the cancer cell lines (**A**) HeLa, (**B**) PC-3, and (**C**) on the normal cell line normal cell L-929 of the sorghum lignin extracts without treatment (LCS) and with the different treatments (L1). Lignin treatment time/amplitude—0/20; (L2) Lignin treatment time/amplitude—10/18; (L3) Lignin treatment time/amplitude—10/14. Samples were used in concentrations ranging from 0–800 µg/mL. The control was DMSO solvent. The data are from three independent experiments. Mean ± standard deviation. Different letters indicate significant difference (*p* < 0.05).

**Figure 8 metabolites-13-00394-f008:**
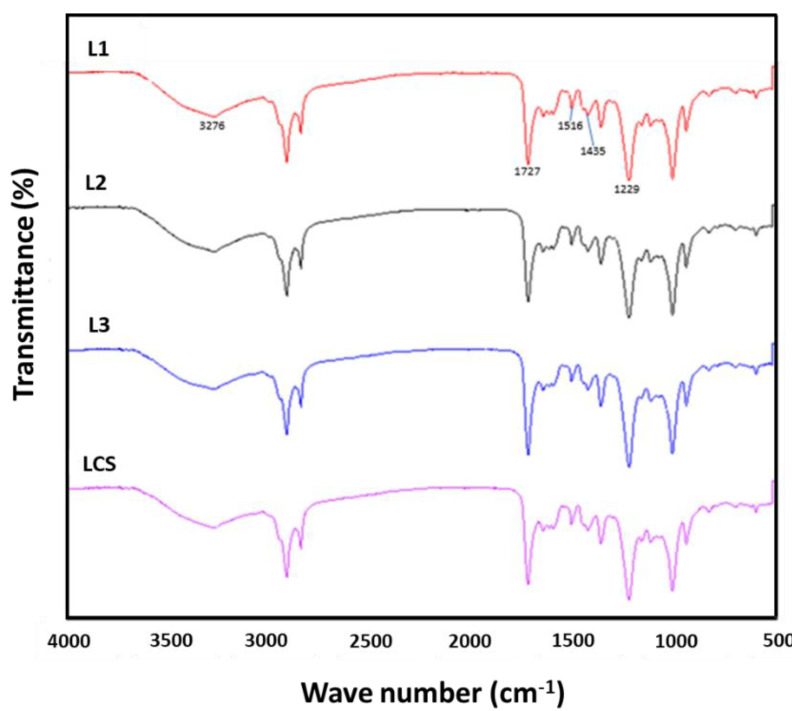
Infrared Fourier Transform (FTIR) of lignin from sorghum Organosolv without (LCS) and with applying high-power ultrasonic pulses. (L1) Lignin treatment time/amplitude—10/20. (L2) Lignin treatment time/amplitude—10/18. (L3) Lignin treatment time/amplitude—10/14.

**Figure 9 metabolites-13-00394-f009:**
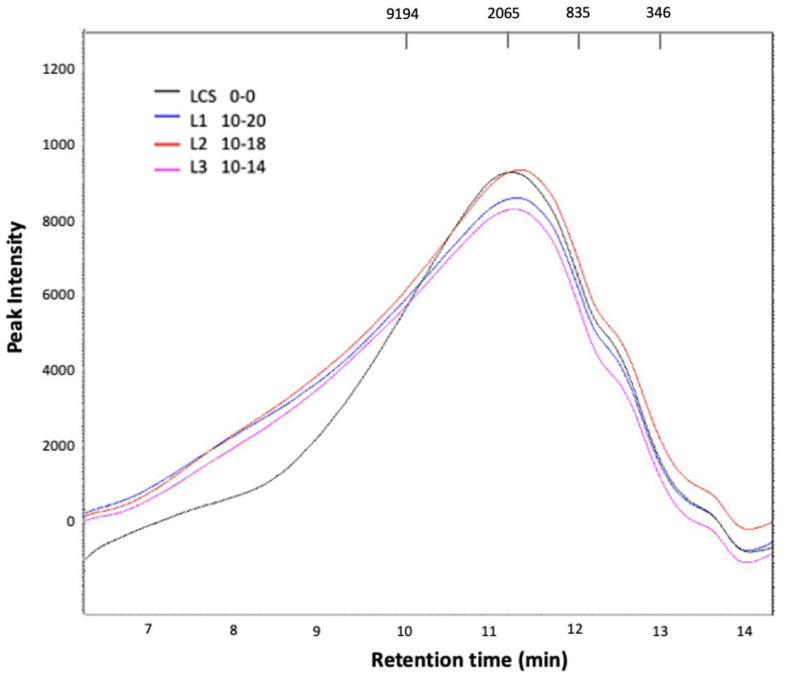
High-performance liquid chromatography of lignin from sorghum organosolv without LCS and by applying high-power ultrasonic pulses. (L1) Lignin treatment time/amplitude—10/20. (L2) Lignin treatment time/amplitude—10/18. (L3) Lignin treatment time/amplitude—10/14. The molecular weights of the standards are shown at the top (g/mol).

**Table 1 metabolites-13-00394-t001:** Composite Central Design to optimize the best conditions for obtaining higher antioxidant capacity and total phenols in sorghum lignin.

Pattern Run	Time (min)	Amplitude
1 −−	0	0
2 −−	0	0
3 a0	0	20
4 a0	0	20
5 −+	0	20
6 −+	0	20
7 0a	5	0
8 0a	5	0
9 OO	5	10
10 OO	5	10
11 OO	5	10
12 OO	5	10
13 0A	5	20
14 0A	5	20
15 +−	10	0
16 +−	10	0
17 A0	10	10
18 A0	10	10
19 ++	10	20
20 ++	10	20

**Table 2 metabolites-13-00394-t002:** Lignin yield from sorghum residues by the Organosolv method.

Extraction	Initial Weight (g)	Final Weight (g)	Sample Yield (g/g)	Yield (%)
1	30	0.6155	0.02	4.39
2	30	0.7493	0.024	5.35
3	30	0.8448	0.028	6.03
4	30	0.8975	0.029	6.41
5	30	1.0156	0.033	7.25
6	30	1.0427	0.034	7.44
7	30	1.2313	0.037	8.79
8	30	1.0645	0.035	7.46
9	30	1.1309	0.037	8.07
10	30	1.159	0.038	8.27
11	30	1.2762	0.042	9.11
12	30	1.2743	0.042	9.1
13	30	1.1055	0.036	7.89
Average	30	1.0313 ± 0.203	0.0334 ± 0.006	7.35 ± 1.45

**Table 3 metabolites-13-00394-t003:** Matrix of the central design composed of the different treatments with and without high-power ultrasonic pulses in sorghum lignin.

min	Amplitude (%)	DPPH	ABTS	FRAP	Phenols	Stand. Dev.	Stand. Dev.	Stand. Dev.	Stand. Dev.
		μmol TE/g	μmol TE/g	μmol TE/g	mg GAE/g	DPPH	ABTS	FRAP	Phenols
0	0	208.23	236.48	1023.96	395.98	29.37	53.60	68.84	58.53
0	0	205.52	236.48	1023.96	395.98	20.58	53.60	68.84	58.53
0	20	208.23	236.48	1023.96	395.98	29.37	53.60	68.84	58.53
0	20	205.52	236.48	1023.96	395.98	20.58	53.60	68.84	58.53
0	20	208.23	236.48	1023.96	395.98	29.37	53.60	68.84	58.53
0	20	205.52	236.48	1023.96	395.98	20.58	53.60	68.84	58.53
5	0	208.23	236.48	1023.96	395.98	29.37	53.60	68.84	58.53
5	0	205.52	236.48	1023.96	395.98	20.58	53.60	68.84	58.53
5	10	211.72	243.95	1084.66	428.28	61.45	56.88	97.33	122.77
5	10	189.45	243.95	1084.66	428.28	14.90	56.88	97.33	122.77
5	10	211.72	243.95	1084.66	428.28	61.45	56.88	97.33	122.77
5	10	189.45	243.95	1084.66	428.28	14.90	56.88	97.33	122.77
5	20	217.33	257.29	1624.60	467.84	31.94	20.18	289.17	54.10
5	20	191.00	257.29	1624.60	467.84	23.64	20.18	289.17	54.10
10	0	208.23	236.48	1023.96	395.98	29.37	53.60	68.84	58.53
10	0	205.52	236.48	1023.96	395.98	20.58	53.60	68.84	58.53
10	10	266.32	273.25	1386.58	483.37	121.50	46.80	294.03	100.84
10	10	169.51	273.25	1386.58	483.37	41.15	46.80	294.03	100.84
10	20	227.40	257.09	1694.88	420.52	37.39	53.87	193.39	26.22
10	20	210.75	257.09	1694.88	420.52	64.82	53.87	193.39	26.22

**Table 4 metabolites-13-00394-t004:** Optimized conditions of high-power pulses on antioxidant capacity and phenolic content in sorghum lignin.

Optimization Results							
DPPH		ABTS		FRAP		Total phenols	
Time (min)	Amplitude (%)	Time (min)	Amplitude (%)	Time (min)	Amplitude (%)	Time (min)	Amplitude (%)
10	20	10	18	10	20	10	14

**Table 5 metabolites-13-00394-t005:** IC_50_ of the lignin samples in the different cell lines.

Cellular Line	LCS	L1	L2	L3
HeLa	252.07 ± 28.56	284.90 ± 23.84	253.92 ± 3.98	253.14 ± 10.05
PC-3	126.52 ± 4.83	171.85 ± 18.91	196.06 ± 12.47	209.28 ± 12.66
L-929	ND	ND	ND	ND

ND (not determined). LCS Lignin extracts from sorghum without treatment (time/amplitude—0/0). L1 Lignin treatment time/amplitude—10/20. L2 Lignin treatment time/amplitude—10/18. L3 Lignin treatment time/amplitude—10/14. IC50 (Mean Inhibitory Concentration; µg/mL) Values for sorghum lignin extracts (µg/mL) represent the mean of at least three independent experiments ± standard deviation. The data are from three independent experiments.

**Table 6 metabolites-13-00394-t006:** Main absorption bands found in the FTIR spectra of lignin organosolv from sorghum.

Wave Number (cm^−1^)	Description
3276	O–H tension vibration of alcohol and phenol groups
1655	C=O tension vibration
1605	C=O stretching of the aromatic ring
1513	Syringile ring vibration and C–O tension vibration
1470	Guayacil ring vibration and C–O tension vibration
1295	C–H stretch of guayacil units (G)
1167	Units p-hydroxyphenyl (H), guaiacil G and syringyl (S)
1122–1056	C–H bond strain in the aromatic ring
947–793	C–H vibration deformation in associated aromatic rings Bending C–H of the syringil units. p-substituted phenolics

**Table 7 metabolites-13-00394-t007:** Assignment of molecular weights of lignin organosolv from sorghum with and without application of high-power pulses. t_R_, Retention time at maximum peak intensity; Mw, Molecular weight. Lignin treatment time/amplitude: (LCS) 0/0; (L1) 10/20; (L2) 10/18; (L3) 10/14.

Sample	t_R_ (min)	Mw (Da)
LCS	11.246	2401.19
L1	11.311	2334.20
L2	11.354	2130.17
L3	11.282	2307.29

## Data Availability

The original contributions data presented in this research are included in the article; further inquiries can be directed to the corresponding authors. The data are not publicly available due to the fact that the author of correspondence keeps control of its diffusion by regulations of the University of Sonora, however, the information can be requested without problem from the people who require it.

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
