# Peer review of "Antioxidant, Antibacterial, Anti-Inflammatory, and Antiproliferative Activity of Sorghum Lignin (Sorghum bicolor) Treated with Ultrasonic Pulses"

_metabolites, 2023, doi:10.3390/metabo13030394_

Round 1

Reviewer 1 Report

Antioxidant, antibacterial, anti-inflammatory, and antiprolifer-2 ative activity of sorghum lignin (Sorghum bicolor) treated with 3 ultrasonic pulses.  

 The article is well written with necessary information. However, in some points related to the mechanisms exerted by lignin, it is not clear. In this experiment, you are trying to use the ultrasonic cavitation treatment to assess its effect on the antioxidant, antibacterial, anti-inflammatory, and antiproliferative activity of sorghum lignin. The ultrasonic treatment is also used in depolymerizing lignin to obtain its derivatives such as vanillin, ferulic acid, syringic acid, eugenol, and other monomeric compounds. These compounds are known to have similar effects, so if you briefly emphasize these monomers, it could be more helpful for the readers.

 On lines 229-231, you are discussing the "tetrazolium sales reduction method." Please check if it is written correctly as "tetrazolium salt reduction method" and if it has a major effect on the assay. Also, briefly describe the modification you used, if any

The article is innovative, and experimental studies have shown a wide pharmaceutical application of lignin. There are some minor errors that the author needs to focus on.

Line 79: what is the solid to liquid ratio. it's mentioned 20:1 (ml/g). Author needs to clarify the solid to liquid ratio clearly.

Line 280 removal or extraction of lignin from lignocellulosic biomass. Don't write carbohydrates.

Line 283: instead of cell write cellulose.

It will be better for the author if lignocellulosic structure can be explained and why is it important to extract lignin?

The author should revise carefully for the grammatical errors present in the manuscript.

Author Response

Thanks to the reviewer for his excellent comments to increase the quality of the article.

  1.  The article is well written with necessary information. However, in some points related to the mechanisms exerted by lignin, it is not clear. In this experiment, you are trying to use the ultrasonic cavitation treatment to assess its effect on the antioxidant, antibacterial, anti-inflammatory, and antiproliferative activity of sorghum lignin. The ultrasonic treatment is also used in depolymerizing lignin to obtain its derivatives such as vanillin, ferulic acid, syringic acid, eugenol, and other monomeric compounds. These compounds are known to have similar effects, so if you briefly emphasize these monomers, it could be more helpful for the readers.

Response 1. This information was added in the introduction part (Lines 50-58).

  1.  On lines 229-231, you are discussing the "tetrazolium sales reduction method." Please check if it is written correctly as "tetrazolium salt reduction method" and if it has a major effect on the assay. Also, briefly describe the modification you used, if any.

Response 2. Thank you for your observation. The name of the method was corrected. Please see line 249. The MTT assay was performed according to the method described by Mossman (1983) with slight modifications made by Hernández et al., (2007). The modified method was described. Please see lines 245-258. 

The article is innovative, and experimental studies have shown a wide pharmaceutical application of lignin. There are some minor errors that the author needs to focus on.

  1. Line 79: what is the solid to liquid ratio. it's mentioned 20:1 (ml/g). Author needs to clarify the solid to liquid ratio clearly.

Response 3: This information was corrected. Line 95.

  1. Line 280 removal or extraction of lignin from lignocellulosic biomass. Don't write carbohydrates.

Response 4. The information was corrected.  Line 292.

  1. Line 283: instead of cell write cellulose.

Response 5: This word was changed. Line 294.

  1. It will be better for the author if lignocellulosic structure can be explained and why is it important to extract lignin?

Response 6. Lignin is the second most abundant polymer on Earth, behind cellulose, and is an affordable renewable resource with potential industrial use. However, the structural definition of lignin has never been as clear as that of other natural polymers, such as cellulose and proteins, due to the complexity that affects its isolation, compositional analysis, and structural characterization. The problem of a precise definition for lignin is associated with the nature of its multiple structural units, which do not usually repeat themselves regularly since the composition and structure of lignin vary depending on its origin, and the method of extraction or isolation used [3].

Notably, the only non-polysaccharide fiber known chemically are lignins, which are phenolic polymers [4]. In general, lignins are copolymers derived primarily from three basic phenylpropane-monomeric units (monolignols): p-coumaryl alcohol, coniferyl al-cohol, and sinapyl alcohol. These monolignols react in the cell wall through oxidation reactions catalyzed by peroxidases (radical intermediates) to finally form lignin polymers. The distribution percentages of the main monolignols depend on the type of plant. Hydrolysis and oxidation of lignin at high pressures and temperatures produce low molecular weight compounds. These compounds represent a variety of high-added-value chemicals, the most important being a group of phenolic compounds that include vanillin, ferulic acid, syringic acid, eugenol, cresols, catechols, and guaiacol, etc., among others [3]. Phenolics are compounds considered antioxidants (they delay or prevent oxidation) that neutralize free radicals that cause cell damage and the development of chron-ic-degenerative diseases [5]. Evidence has shown that the lignin extracted from sorghum residues has a high antioxidant capacity (75% inhibition of the DPPH radical) [6]. The antioxidant capacity, on many occasions, has also been related to other biological activities such as antimicrobial and anticancer [7]. However, those activities have not been de-termined in sorghum residues. (Lines 42-64).

  1. The author should revise carefully for the grammatical errors present in the manuscript.

Response 7. The writing in English was reviewed by a language expert.

Reviewer 2 Report

Please give the overall goal of the manuscript. Explain why you have chosen these bacteria and cell lines.

Line 23: What is the AB group?

Line 133: Write “total phenolic content”

Line 136: Write “sodium carbonate”

Line 160: From where did you receive the blood samples?

Lind 171: Write “… in …”

Line 226: Write Dulbecco

Line 230: Write the manufacturer of the MTT kit.

Line 240: Do you mean 50 µL??

Line 245: I think you have to delete “-655 nm”? and add the city and country of the device.

Line 252: Write the manufacturer, city, and country

Table 2: What do you mean with “01.0313” ??? “1.0313”??

Figure 1: Should be removed to line 305.

Figures 2, 3, and 4: The concentration is difficult to read.

In figure 4: What about AB+??

Line 515-518: Delete. Everyone who works with HPLC knows that.

Figure 9: is an HPLC chromatogram??? No! It looks like spectra? What means the different lines mean? Which is the sample? Which is standard?

Table 7: Give more information. Explain the abbreviations? What can I see at 11 and 15 min?

Author Response

Thanks to the reviewer for his excellent comments to increase the quality of the article.

  1. Please give the overall goal of the manuscript. Explain why you have chosen these bacteria and cell lines.

Response 1. The objective of this research was to evaluate the effect of high-power pulses on the an-tioxidant, antibacterial, and anticancer activity and the structure of sorghum lignin. (Lines 79-80).

An explanation of why these bacteria and cells lines have been chosen was included. Please see  lines 437-441  and lines 456-458.

  1. Line 23: What is the AB group?

Response 2. The type of AB blood group. This was corrected in line 23.

  1. Line 133: Write “total phenolic content”

Response 3. This was corrected. Line 148.

  1. Line 136: Write “sodium carbonate”

Response 4. This was corrected. Line 151.

  1. Line 160: From where did you receive the blood samples?

Response 5 Blood samples were obtained from healthy volunteers older than 18 with prior informed consent.This was added in lines 173-174.  The ethical part was explained in the corresponding section.

  1. Lind 171: Write “… in …”

Response 6. This was corrected. Line 186.

  1. Line 226: Write Dulbecco

Response 7. This was corrected. Line 242.

  1. Line 230: Write the manufacturer of the MTT kit.

Response 8. The manufacturer of the MTT reagent was included. Please see line 247-247.

  1. Line 240: Do you mean 50 µL??

Response 9. The volume units were corrected. Please see line 250.

  1. Line 245: I think you have to delete “-655 nm”? and add the city and country of the device.

 Response 10. Thank you very much for your observation. The wavelengths were corrected, and the city and country of the device were included. Please see lines 261-262.

  1. Line 252: Write the manufacturer, city, and country

Response 11. The information was added as suggested. Line 256.  

  1. Table 2: What do you mean with “01.0313” ??? “1.0313”??

Response 12. The correct is 1.0313. This was corrected.

  1. Figure 1: Should be removed to line 305.

Response 13. Figure 1 was removed as the reviewer suggested.

  1. Figures 2, 3, and 4: The concentration is difficult to read.

Response 14. The concentrations became more readable.

  1. In figure 4: What about AB+??

Response 15. For this technique, the AB blood type could not be used due to its scarcity. This part was written in the methodology section (Lines 202-203).

  1. Line 515-518: Delete. Everyone who works with HPLC knows that.

Response 16. The information was deleted as suggested.

  1. Figure 9: is an HPLC chromatogram??? No! It looks like spectra? What means the different lines mean? Which is the sample? Which is standard?

Response 17. The axis of the chromatogram in Figure 9 was changed from RIU (Refractive Index Units) to Peak Intensity. Also, information was added to the figure title: “High-performance liquid chromatography of lignin from sorghum organosolv without (LCS) and with applying high-power ultrasonic pulses. (L1) Lignin treatment time/amplitude -10/20. (L2) Lignin treatment time/amplitude-10/18. (L3) Lignin treat-ment time/amplitude-10/14”. The molecular weights of standards are shown at the top (g/mol). Likewise, the text was modified in line 550.

  1. Table 7: Give more information. Explain the abbreviations? What can I see at 11 and 15 min?

Response 18. The Table 7 was modified. Also, and explanation of the abbreviations was added: “tR, Retention time at maximum peak intensity; Mw, Molecular weight. Lignin treatment time/amplitude: (LCS) 0/0; (L1) 10/20; (L2) 10/18; (L3) 10/14.

Round 2

Reviewer 2 Report

no further comments